# The Impact of Entrepreneurial Orientation on the Sustainable Innovation Capabilities of New Ventures: From the Perspective of Ambidextrous Learning

**Xihua Yu [1], Ning Cao [2,*] and Hao Ren [1]**

1   School of Economics and Management, Tongji University, Shanghai 200092, China; 1310396@tongji.edu.cn (X.Y.); renhao@tongji.edu.cn (H.R.)
2   School of Business, Shanghai Dianji University, Shanghai 201306, China
*   Correspondence: caon@sdju.edu.cn

**Abstract:** Amid changes in the business environment and increased competition, sustainable innovation has become the key for new ventures to survive and develop. Innovation capability is considered to be closely related to entrepreneurial orientation and organizational learning. However, there was no in-depth analysis of sustainable innovation capability from an ambidextrous perspective to distinguish from traditional single-episode innovation and no systematic empirical study to explore the internal relationship among the three factors mentioned above in a new venture scenario. Employing the ambidextrous perspective, this paper explored the impact of entrepreneurial orientation on the sustainable innovation capabilities of new ventures and examined the mediating effect of ambidextrous learning. Using the multisource data of 279 new ventures from China, this paper empirically tested the research hypothesis. The results showed that: (1) The three dimensions of entrepreneurial orientation have a significant positive impact on the sustainable innovation capability of new ventures; (2) Ambidextrous learning partially mediates the relationship between entrepreneurial orientation and the sustainable innovation capability of new ventures; (3) Both the equilibrium and interaction effects of ambidextrous learning positively affect the sustainable innovation capability of new ventures. This study clarified the relationship between entrepreneurial orientation and the sustainable innovation capability of new ventures, emphasized the equilibrium and interaction in ambidextrous learning, and provided theoretical support for new ventures to form and develop sustainable innovation capability.

**Keywords:** new venture; sustainable innovation capabilities; entrepreneurial orientation; ambidextrous learning

## 1. Introduction

New ventures are generally built on new "ideas" and grow through innovation. However, the income generated by innovation inevitably leads to imitation, user preferences and government policies change with the emergence of new technologies, and the enterprise's inimitable resources, capabilities, and first-mover advantage gradually diminish, resulting in the enterprise losing its competitive advantage [1]. Research indicates that approximately 50% of new ventures survive for just one and a half years, while fewer than 30% survive for more than six years [2]. Many new ventures face problems such as business collapse, unsustainable innovation activities, short life cycles, and a lack of corporate vitality. In a static sense, a single innovation episode can no longer sufficiently serve as a company's sustainable competitive advantage. Sustainable innovation and improvement are increasingly necessary and important for the survival of a company, particularly in terms of enhancing technology, building influence, and satisfying the needs of different customers quickly and efficiently [3].

New ventures are also affected by the "disadvantage of new entry" [2,4] and suffer from inherent defects such as insufficient internal resources, financing difficulties, low legitimacy, and a lack of social network relationships [5]. Amid high uncertainty and survival pressure, sustainable innovation is key to new ventures enhancing their vitality, survival rate, and entrepreneurial success rate in a dynamic environment. Accordingly, sustainable innovation capability (SIC) determines the future development path and competitive advantage of new ventures. Therefore, building SIC and enhancing competitive advantage are important measures by which new ventures can improve their entrepreneurial success and implement innovation-driven strategies.

In the literature on sustainable innovation, scholars typically define "sustainable" in terms of linear time, emphasize "sustainable innovation over a long period" and "continuous improvement" superficially, or equate it with traditional innovation [6–8]. Most scholars agree that sustainable innovation is "the continuous process of perceiving, exploring and learning, which enables enterprises and business organizations to innovate new procedures in business organizations, and new markets and improved products and services" [7]. This linear time perspective makes it difficult to define the operational and measurable dimensions of SIC as a concept, and related research largely comprises theoretical inquiries, with a marked lack of strictly empirical studies [9,10]. According to paradox theory, the tension between two sides of a paradox (called Yin and Yang in Chinese) is interrelated and contradictory, the ambidextrous relationship coexists for a long time, and provides theoretical support for "sustainability" [11–14]. Therefore, ambidextrous relationship could become an important perspective to solve the problems above.

In addition, new ventures have different ways of understanding sustainable innovation, resulting in different approaches to integrate sustainable innovation into their business [15]. Despite abundant research on the performance, behavior, activities, antecedents, and consequences of sustainable innovation as a single dimensional concept [16,17], they lack research in a new venture scenario. Few scholars have analyzed what SIC of new ventures is or how it is achieved.

Among empirical studies on the influencing factors of innovation capability, scholars have identified entrepreneurial orientation (EO) as the source of an enterprise's dynamic capability [18], viewing it as the spirit of organizational innovation, and facilitating the formation and internal enhancement of dynamic capability [19]. Ferreras-Mendez [18] investigated and found that the small- and medium-sized enterprises must possess EO consciousness, proactiveness, and risk tolerance, which will help them develop products and services and create breakthrough technologies. The question is, in employing the ambidextrous perspective, is this still appropriate in the new ventures scenario? Meanwhile, the relationship between EO and firm performance has been trapped in constructing complex quantitative models by constantly adding moderating and mediating variables. Regarding the key transformation path from EO to innovation capability, some scholars believed that EO has an indirect impact through other variables, such as business model, absorptive capability, and organizational learning [18–20], rather than a direct impact. Some regarded ambidextrous learning as a bridge to realize the reconstruction of sustainable innovation [7]. So with defining the SIC from an ambidextrous perspective, what role does ambidextrous learning play between EO and the SIC in a new venture scenario? What about the equilibrium and interaction effects of ambidextrous learning?

Given the foregoing, this empirical study defines the SIC of new ventures from an ambidextrous perspective, explores the formation mechanism, and constructs and examines a model of the relationship between EO and ambidextrous learning and the SIC of new ventures. Based on this model, this study answers the following research questions: (1) Can new ventures enhance SIC through entrepreneurial orientation? (2) What role does ambidextrous learning play between EO and the SIC of new ventures? (3) Can new ventures enhance SIC through the equilibrium and interaction effects of ambidextrous learning?

## 2. Theoretical Background and Research Hypotheses

Centering on the above issues, this part of the study is mainly dedicated to the presentation of the related literature as guided by the research theory.

### 2.1. Entrepreneurial Orientation

EO reflects an enterprise's commitment, ability, and desire to engage in entrepreneurial behavior. This kind of "strategic posture" indicates an enterprise's willingness to take business-related risks, support change and innovation, and actively pursue competitive advantages [21]. Understood as an overall competition orientation, EO comprises three dimensions: innovativeness, proactiveness, and risk-taking [22]. Innovativeness refers to an enterprise's willingness to develop new products and services through creative thinking, reflecting the significance of pursuing "new entry" opportunities. Proactiveness refers to an enterprise's efforts to identify and seize opportunities. Enterprises tend to act by predicting future market demands, including launching new products or services and introducing new technologies ahead of their competitors. Finally, risk-taking refers to an enterprise's tendencies when making decisions and acting when results are uncertain, its willingness to invest resources in unknown or high-risk businesses, and its tolerance of failure [10,22].

### 2.2. Ambidextrous Learning

"Ambidextrous" means that a management entity can successfully achieve two closely related things that are often difficult to achieve together [23–25]. "Ambidextrous elements" imply the paradox of coordination, correlation, and contradiction on the one hand, and the mutual promotion of dialectical unity on the other. The contradiction between the two sides should be grasped with both hands to maintain their dynamic balance.

Ambidextrous learning means that an organization simultaneously adopts interrelated but contradictory learning forms, such as exploratory and exploitative learning. Exploratory learning centers on experimentation with new solutions to obtain new knowledge and resources. More specifically, it refers to an enterprise's search and use of experience and knowledge beyond its current market and product experience and initiative to experiment despite uncertain returns. Meanwhile, exploitative learning focuses on refining and expanding existing capabilities, technologies, and patterns to utilize existing knowledge and resources. In this respect, enterprises focus on acquiring relevant market and product information, improving and integrating existing knowledge and skills, and constantly improving productivity and efficiency by updating their knowledge with predictable returns [26,27].

### 2.3. The SIC of New Ventures

A new venture was defined as "an entrepreneur who takes advantage of a business opportunity in the market and establishes a legal entity through the integration of resources that focuses on providing products and services with the goal of growth and sustained profitability" [2,4,5]. Today's commercial, social, and technological environment is changing rapidly, the life cycle of enterprises is shortening by the day, and the characteristics of enterprises are becoming increasingly diverse. The enterprise age cannot be used as a single standard of measurement. This study defines new ventures as enterprises that have not yet entered the mature stage, which means the key financial indicator (sales proceeds) has not yet reached a stable state [28]. Compared with mature enterprises, new ventures are affected by the "disadvantage of new entry" and face various difficulties, such as insufficient internal resources, financing difficulties, low legitimacy, lack of social network relationships, an immature and unstable organizational structure [2], as well as higher uncertainty and survival pressure. However, new ventures have the advantage of being "small but beautiful" which allows them to break through various framework constraints and exhibit greater innovativeness. For instance, flexible organizational culture and systems can support the rapid iteration of organizational structure, team members are full of entrepreneurial passion, and technological innovation ability grows faster [29].

Elucidating a form of tension that is both contradictory and complementary, Lewis, an authoritative scholar of western paradox theory, defined paradox as the "persistent contradiction between interdependent elements" and its essence as "elements of long-term coexistence of interrelated and conflicting simultaneity" [12]. According to paradox theory, the tension between the two sides of a paradox is interrelated and contradictory, the binary relationship coexists for a long time, and sustainable performance depends on the effectiveness of paradox management. In Chinese Taoism, everything in nature and social life can be divided into two conflicting and complementary components, namely, Yin and Yang, with the state of all things explained by the balance of any change in the two components [13]. The balance between Yin and Yang is a fundamental law for the long-term development of things, which constantly change in cycles of balance, imbalance, and rebalancing. The paradoxical tension and dynamic balance between two seemingly opposing sides or Yin and Yang are integral for sustainable development. Such interrelated and contradictory tension in a long-term binary relationship provides theoretical support for sustainability. Therefore, the fundamental difference between sustainable and traditional innovation modes lies in the tensions generated by the innovation paradox. Produced by conflicting demands and activities, tension is at the core of innovation [14]. Sustainable innovation is produced by dialectical unity and the dynamic balance of innovation tensions. The conflicts and contradictions between the two sides of innovation tension are interdependent and influence each other. Accordingly, sustainable innovation requires enterprises to attend to and maintain the dynamic balance of both sides of innovation tension (e.g., "creation" and "implementation", "exploration" and "utilization") [15].

From a knowledge perspective, enterprise innovation capability emphasizes the ability to form new knowledge or technology through knowledge search, integration, diffusion, and transformation [30]. However, an enterprise's innovation capability is not the equivalent of knowledge; rather, it emerges from the dynamic interactions of knowledge. From a resource perspective, together with the resources obtained from internal and external inputs, an enterprise's innovation capability is the recessive determinant of innovation performance [31]. From an efficiency perspective, enterprise innovation capability is a kind of efficiency, namely, converting resource input into the target output [32]. Input–output efficiency represents the ability set required for the creation, application, and transformation of new ventures' knowledge to create market value and obtain sustainable competitive advantages. Finally, from a process perspective, innovation capability is not a static state but a dynamic spiraling process involving the interaction among environmental support, innovation potential, innovation input, organizational operation, and output capabilities [33,34].

In summary, SIC in new ventures involves taking market opportunities, both existing and potential as the starting point, sustainable innovation as the goal, self-organization and self-motivation as the driving force, and organizational learning as the basis. New ventures use unique resources, knowledge, and skills to innovatively integrate various production factors and maintain the dynamic balance of innovation tension throughout the spiraling development process. From the ambidextrous perspective, a remarkable characteristic of the SIC of a new venture is the paradox of stable reuse and capability iteration. On the one hand, a new enterprise needs a stable and profitable organizational path to survive. Structured conventions are the basis of the stable reuse of enterprise behavior and a necessary condition for the survival of new enterprises [35]. To ensure the reproducibility of performance, new ventures must continuously reuse existing capabilities to achieve stable revenue [36]. However, this process intensifies the capability and behavioral rigidity derived from organizational conventions. As a result, innovation capability is locked and falls into the capability trap. The ambidextrous paradox of the SIC of new ventures comprises the significant contradiction between the stability and development of capability. The two conflicts promote each other, both challenging and revitalizing new ventures.

### 2.4. Entrepreneurial Orientation, Ambidextrous Learning, and SIC

The influence path of EO on the SIC of new ventures involves innovating products and services, winning first-mover advantage in competition using the preemptive method, and implementing calculated risk strategies to obtain greater risk returns [37]. First, as new ventures are restricted by "new entry disadvantages"—such as lack of resources, immature market environment, and imperfect infrastructure—innovativeness makes them more likely to accept new ideas and methods, utilize resources creatively, and make adjustments and adaptations in response to changes and trends in the business environment [38]. This process of constantly accepting and adopting new ideas, methods, and thinking in pursuit of "new entry" opportunities promotes mutual learning among various departments and levels, the exchange of creative thinking, and internal and external resource integration, and accelerates the flow of information and product updating. Therefore, new ventures open up new markets and launch new products or services faster than their competitors. While this process avoids the rigidity of capability, it promotes the formation and development of the SIC of new enterprises.

Second, proactive start-ups scan the external environment and identify business opportunities in the market. Entrepreneurial companies often create, identify, and develop favorable markets and opportunities ahead of their competitors or more easily [39]. This can help companies gain sustained dominance, create first-mover product or service advantages, seize market segments, and control market prices [40]. A pioneering venture takes preemptive action in launching new products or services and introducing new technologies. In this process, the new venture will reintegrate, create, apply, and transform knowledge, thereby realizing its sustainable innovation ability.

Finally, risk-taking leads to higher organizational commitment and R&D investment. When new ventures encounter uncertainty in terms of innovation inputs and outputs, enterprises with high risk-taking tendencies are more willing to invest resources in unknown businesses and are more tolerant of failure [19]. Characteristics such as "small size and flexibility", "light historical burden", and "low impact of failure" allow new ventures to make decisions quickly and turn negative impacts, such as lack of resources or inadequate infrastructure, into advantages. In the process of taking risks, new ventures seize opportunities and take preemptive actions, thereby reintegrating, creating, implementing, and transforming knowledge and realizing their sustainable innovation capabilities.

Therefore, based on the foregoing, this study proposes the following hypothesis:

**H1.** *EO has a positive influence on the SIC of new ventures.*

New innovative ventures show a greater willingness to develop new products, services, and technologies, driving them to research, analyze, and process highly experimental and risky information, pursue novel information and ideas, and seek new solutions to problems that facilitate the transfer and transformation of knowledge [41]. Innovativeness helps new ventures enter new fields, change their situation, and explore new areas [42]. In addition to promoting creative thinking and learning, innovativeness drives exploratory learning to transform innovative ideas into new products or services.

Proactive new ventures want to be front-runners, proactively perceive and anticipate changes in the environment, and identify new opportunities. This promotes enterprises to enter the training state and fosters a high level of learning self-efficacy and development of learning goals. This predisposition sees enterprises focus on new areas of experimentation, markets, and technologies, gather novel information and ideas beyond the existing market and technology experience, and discover new business opportunities through exploratory learning. Pioneering behavior motivates new ventures to engage in creative activities [43], find innovative solutions to problems, and develop abstract creative approaches. In addition, when they identify new business opportunities and market gaps, preemptive action drives organizational learning insofar as it requires startups to have certain skills and knowledge and/or identify the gap between their current state and the knowledge and skills they need to achieve their goals [44].

New ventures with high risks are willing to assume tolerable risks and manage risks or uncertainties beyond expectations using relevant management tools. In this regard, employees need to exchange tacit knowledge, solve problems, and avoid increasing risks through organizational learning [45]. Organizational learning can help new ventures overcome uncertainty by enhancing their ability to access market information, which facilitates product development and commercialization, thereby facilitating the solving of problems and reduction of uncertainty [46]. At the motivational level, companies with a high propensity for risk-taking are more willing to invest resources in unknown businesses and tolerate failure, creating a more open and accepting atmosphere and willingness to learn from multiple perspectives and experiences [47].

Therefore, this study proposes the following hypotheses:

**H2.** *EO positively influences exploratory learning in new ventures.*

**H2a.** *Innovativeness positively influences exploratory learning in new ventures.*

**H2b.** *Proactiveness positively influences exploratory learning in new ventures.*

**H2c.** *Risk-taking positively influences exploratory learning in new ventures.*

To ensure the successful launch of their innovations, startups conduct exploitative learning, research their customers and competitors, identify productivity technologies and other information, and seek solutions to general issues. Moreover, the crisis of a new-entry disadvantage in terms of resources prompts ventures to focus on extreme and creative uses of existing resources. New ventures are also more receptive to new ideas and methods, more creative in using resources, and more open to making adjustments in response to changes and trends in the business environment [45]. The extreme and creative use of resources results in new ventures building on and strengthening their existing knowledge, skills, processes, structures, and designs, enhancing the performance of existing products and services, and improving the efficiency of existing sales channels [44].

In addition to the foregoing, proactiveness drives new ventures to continuously analyze their business environment, collect information on customers and competitors, address any technical gaps in existing products or services, evaluate their current situation and possible directions for exploration through exploitative learning, and face new entry disadvantages [46]. To overcome the disadvantage of new entry, proactiveness promotes new enterprises to improve production efficiency, broaden the existing knowledge, skills, processes, and structures of the organization, improve the existing design, enhance the performance of products and services, and raise the efficiency of existing sales channels. In other words, the existing advantages of new ventures are consolidated through exploitative learning.

When facing uncertainty, new ventures can respond quickly because they are small and flexible, free of historical baggage, and not overly impacted by failure. At the same time, they search for ways to solve existing problems and strengthen their knowledge, skills, products, and designs to improve their performance, harness advantages, and reduce the risk of uncertainty.

Accordingly, this study proposes the following hypothesis:

**H3.** *EO positively influences exploitative learning in new ventures.*

**H3a.** *Innovativeness positively influences exploitative learning in new ventures.*

**H3b.** *Proactiveness positively influences exploitative learning in new ventures.*

**H3c.** *Risk-taking positively influences exploitative learning in new ventures.*

Exploratory learning emphasizes the acquisition of relevant knowledge and information in new fields, influencing the ability to perceive and identify opportunities for new ventures and enabling their entrance into or opening of new markets [48]. Accumulated experience and knowledge enrich organizational knowledge resources and facilitate the development of dynamic capabilities. Exploratory learning influences an organization's

ability to adapt to an uncertain environment and make decisions quickly. Innovation and experimental learning behaviors help organizations identify new technologies, create new opportunities, and launch new products or services faster than their competitors [49]. Exploratory learning also emphasizes experimentation and innovation, breaks existing learning paths and practices, promotes the reorganization and reconstruction of resources, develops new technologies, tracks industry trends, promotes changes in organizational practices through repeated experimentation and knowledge coding, and acquires new knowledge. Consequently, the enterprise can avoid capability rigidity and expand its cognitive boundaries, promoting the improvement and iteration of sustainable innovation capabilities in new ventures.

Exploitative learning emphasizes the understanding and utilization of existing knowledge and resources and helps organizations improve dynamic capabilities by expanding the content and depth of their knowledge resources [50], deepening the understanding of organizational processes and practices by emphasizing refinement, efficiency, and execution [25], and facilitating the adjustment of resource allocation and adaptation to changes in the external environment. Consequently, enterprises can better understand market segmentation and competitive situations, as well as acquire and exploit opportunities in a timely manner. Enterprises can better perceive opportunities and threats using the knowledge and experience they accumulate through exploitative learning, update their technology to change the competitive environment [51], create opportunities, and avoid the risks of technological shifts in the industry. A high level of exploitative learning can help new ventures learn and explore knowledge areas, improve operational efficiency, and transform existing knowledge into new products and services. This process can reduce the risk of exploratory learning while satisfying the short-term interests of new ventures.

Therefore, this study proposes the following hypotheses:

**H4a.** *Exploratory learning positively influences the SIC of new ventures and mediates the relationship between EO and the SIC of new ventures.*

**H4b.** *Exploitative learning positively influences the SIC of new ventures and mediates the relationship between EO and the SIC of new ventures.*

Some believe that exploratory and exploitative learning contradict and conflict with each other—a continuum [52] typically called the equilibrium dimension of two elements. Others view these learning approaches as mutually complementary, interrelated, and mutually promoting activities—as orthogonal bodies commonly known as interactive dimensions.

In ambidextrous learning, the dimension of balance reflects how the two behaviors of exploratory and exploitative learning compete for resources to achieve their own goals [53]. Maintaining an appropriate balance between exploring new possibilities and exploiting old certainties is a primary factor in organizational survival and prosperity. The excessive pursuit of exploitative activities leads to organizational inertia and conservatism. Exploitative learning has a crowding-out effect on exploratory learning, which may be very effective in the short term, but tends to lead to organizational demise in the long term. The excessive pursuit of exploratory activities reduces the efficiency of business operations, hindering economies of scale and learning effects [49]. Accordingly, sustained competitive advantage is rooted in the organization's ability to maintain both short-term efficiency and long-term innovation.

Organizational learning is a continuum of exploratory and exploitative learning, and organizations cannot lean too far one way or the other when maintaining balance in this continuum [48]. Scholars suggest that organizations need strong integration capabilities to reconcile the tensions between the two types of learning behaviors and avoid a decline in performance [54]. New ventures generally use the two learning methods simultaneously. New ventures seek to grow in the long-term balance, choose between the two learning activities according to their strategic goals and resource constraints, and meet both short-term

survival and long-term development needs, steadily building their sustainable innovation capabilities.

Based on the foregoing, this study proposes the following hypothesis:

**H5.** *The equilibrium effect of ambidextrous learning positively influences the SIC of new ventures.*

Meanwhile, the interaction dimension of ambidextrous learning views exploration and exploitation as relatively independent activities that are interconnected and complementary [52]. Exploratory and exploitative learning behaviors need to efficiently coordinate and promote each other in terms of the interaction dimension. Exploratory learning constantly expands the organization's knowledge boundary. Adding a new knowledge domain influences the organization's conception of existing resources and knowledge, helping new ventures enhance their existing operational processes, products, or services, and enhance the exploitation and utilization of existing knowledge resources. Exploratory learning increases the depth of exploitative learning. Exploitative learning emphasizes the comprehensive evaluation of the enterprise's current domain knowledge and internal resources, the decomposition, reorganization, integration, and reuse of such knowledge and resources, and the revitalization of the enterprise's understanding of existing knowledge and behavior. Consequently, the identification, acquisition, reconstruction, and integration of new resource knowledge are more logical [55,56]. From this perspective, the depth, hierarchy, and effectiveness of exploitative learning contribute significantly to exploratory learning. The high-quality development cycle, in which exploratory and exploitative learning complement and promote each other, facilitates the stability, iteration, and improvement of the SIC of new ventures.

Accordingly, this study proposes the following hypothesis:

**H6.** *The interaction effect of ambidextrous learning positively influences the SIC of new ventures.*

Figure 1 summates this study's hypotheses and presents the proposed model, illustrating the hypothetical relationships between EO as an independent variable, the SIC of new ventures as a dependent variable, and ambidextrous learning as a mediating variable.

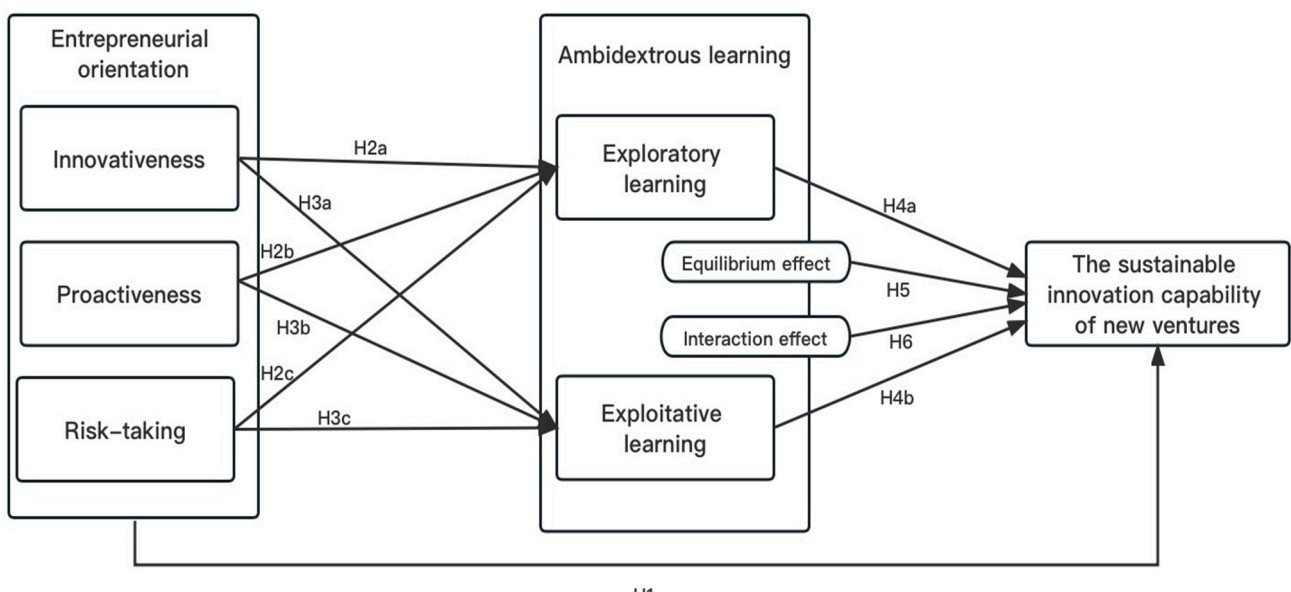

**Figure 1.** The theoretical framework of this research.

## 3. Methodology

To test our hypothesis, we formulated the measurement scale and adopted the method of a large sample questionnaire to obtain sufficient data for the research.

### 3.1. Sample and Data

The selection of the sample was based on three criteria: the enterprise (1) had not entered the mature stage according to enterprise life cycle theory, (2) was engaged in intensive knowledge and technology production with relatively low consumption of material resources, and (3) conducted certain activities such as product development or technological innovation. Generally, the more developed the regional economy, the more mature the manufacturing industry, the better the preferential policies and infrastructure, the higher the quality of talent, the stronger the innovation vitality of the region, and the more new enterprises there are. To ensure the validity of the survey data, we selected several developed provinces and cities which are relatively active in innovation activities as the distribution regions, namely, Jiangsu, Zhejiang, Shanghai, Beijing, Shenzhen, Guangzhou, Tianjin, and Wuhan in China.

First, we conducted pre-research and testing on a small sample, primarily of startup technology companies, over approximately four months in Beijing, Tianjin, and Hebei provinces. Survey targets were middle and senior management with more than three years of work experience and familiarity with the company's strategy, research and innovation, internal learning, and overall operation. It took much time to communicate and interview the interviewees, which mainly included the explanation of the purpose and method of the survey, the explanation of the questionnaire terms and sentences, the confidentiality clause, and so on. A total of 160 questionnaires were distributed in the pre-research phase; 120 were returned, and invalid questionnaires were deleted to produce a total of 81 valid questionnaires for a recovery rate of 75% and an effective rate of 50.62%. Finally, we conducted reliability and exploratory factor analyses of the questionnaire, deleted some questions, and formed a formal questionnaire for the large-scale survey.

Large-scale data collection lasted approximately 12 months. A total of 180 questionnaires were disseminated via telephone, email, and other information channels; of the 97 returned, 45 were valid. An additional 470 questionnaires were distributed through the professional survey website Star, with 142 valid questionnaires collected. A further 156 questionnaires were distributed to in-service students enrolled in EMBA and MBA classes at well-known universities, with 120 valid questionnaires collected. Additionally, 806 questionnaires were distributed to the sample enterprises and 457 questionnaires were returned. The project team then carefully checked and screened the questionnaires. A final total of 279 valid questionnaires was obtained after eliminating questionnaires that were incomplete or had inconsistent or highly similar answers, including excessively "uncertain" answers.

### 3.2. Variable Measurement

This study used authoritative scales from domestic and foreign scholars to collect data and ensure variable reliability and validity of the variables. Variables were measured using a five-point Likert scale ranging from 1 (strongly disagree) to 5 (strongly agree). We examined three dimensions: entrepreneurial orientation, the sustainable innovation capability of new ventures, and ambidextrous learning.

EO was divided into three dimensions—innovativeness, proactiveness, and risk-taking—and measured using 10 items adopted from the literature [40,57]. Innovativeness included four items, such as "Top management highly values R&D, technology leadership, and innovation" and "Over the past three years, it has been normal and natural for the company to iterate over product or service lines". Proactiveness comprised three items, such as "In dealing with competitors, the enterprise acts proactively rather than passively". Finally, risk-taking contained three items, such as "The enterprise prefers high-risk projects with uncertain returns".

Pairwise comparison was adopted to measure the SIC of new ventures [28,58]. This dimension was measured using five items, such as "We can introduce new products or services faster than our competitors" and "We can open up new markets faster than our competitors".

Ambidextrous learning was divided into two dimensions—exploratory learning and exploitative learning—and measured using 10 items adopted from the literature [59]. Exploratory learning included five items, such as "In terms of information collection, the company focuses on acquiring strategic knowledge involving experimentation and high market risk". Exploitative learning similarly comprised five items, including "In terms of information collection, the company focuses on acquiring knowledge to solve existing problems".

This study identified the equilibrium and interaction effects of ambidextrous learning based on He and Wong, Cao et al., Wong et al., and Wang et al. [60–63]. First, we eliminated multicollinearity by standardizing exploratory learning and exploitative learning and then multiplying the two, producing an interaction effect of $X \times Y$ and an equilibrium effect of $1 - |X - Y|/(X + Y)$. The absolute value of the difference between exploratory and exploitative learning was divided by their sum to reveal the degree of imbalance in ambidextrous learning. The degree of equilibrium was obtained by subtracting the degree of imbalance from 1. The larger the value, the broader the difference between exploratory and exploitative learning.

In terms of control variables, factors such as the size and age of new ventures are relevant to the process of improving and developing SIC. Accordingly, we set the size and age of new ventures as control variables measured by the number of employees and operating years, respectively.

## 4. Results

### 4.1. Measurement Model Estimation

Before testing the research hypotheses, we checked the validity and reliability of the measurement model. Based on 81 valid questionnaires collected during the pre-investigation stage and using SPSS24.0 software, we conducted Exploratory Factor Analysis (EFA) and the CITC test on the entrepreneurial orientation, ambidextrous learning, SIC of a new venture's dimension, verified and purified the measurement items of related variables, and subsequently deleted some items [12]. Based on 279 questionnaires collected during the formal investigation stage, we conducted a Confirmatory Factor Analysis (CFA) of the revised scale and adopted Maximum Likelihood Estimation (MLE) to verify the model's coefficient of determination.

As Table 1 shows, the alpha coefficients of innovativeness, proactiveness, risk-taking, exploratory learning, exploitative learning, and the SIC of new ventures are above 0.8 and thus pass the reliability test. Factor loadings are all above 0.5, CR values above 0.7, and AVE values above 0.5, indicating good convergent validity. All fitting values met the judgment criteria, indicating that the CFA model fitted well.

**Table 1.** The validity and reliability of the research measurement model.

| Variables | | Item | Factor Loading | CR | AVE | Goodness of Fit |
|---|---|---|---|---|---|---|
| Entrepreneurial orientation (EO) | Innovativeness ($\alpha$ = 0.846) | IT1<br>IT2<br>IT3<br>IT4 | 0.756<br>0.746<br>0.776<br>0.777 | 0.849 | 0.584 | $X^2/DF$ = 2.594<br>RMSEA = 0.070<br>GFI = 0.958<br>NFI = 0.949<br>TLI = 0.955<br>IFI = 0.968<br>CFI = 0.968 |
| | Proactiveness ($\alpha$ = 0.853) | PT1<br>PT2<br>PT3 | 0.822<br>0.820<br>0.796 | 0.854 | 0.661 | |
| | Risk-taking ($\alpha$ = 0.857) | RT1<br>RT2<br>RT3 | 0.787<br>0.814<br>0.853 | 0.859 | 0.670 | |

**Table 1.** *Cont.*

| Variables | | Item | Factor Loading | CR | AVE | Goodness of Fit |
|---|---|---|---|---|---|---|
| The SIC of new ventures ($\alpha$ = 0.890) | | SIC1 SIC2 SIC3 SIC4 SIC5 | 0.815 0.885 0.875 0.807 0.805 | 0.922 | 0.702 | $X^2$/DF = 2.423 RMSEA = 0.066 GFI = 0.986; NFI = 0.990 TLI = 0.988; IFI = 0.994 CFI = 0.994 |
| Ambidextrous learning (AL) | Exploratory learning ($\alpha$ = 0.895) | EXPR1 EXPR2 EXPR3 EXPR4 EXPR5 | 0.807 0.753 0.809 0.860 0.838 | 0.907 | 0.663 | $X^2$/DF = 1.674 RMSEA = 0.045 GFI = 0.967 NFI = 0.968 TLI = 0.983 IFI = 0.987 CFI = 0.987 |
| | Exploitative learning ($\alpha$ = 0.859) | EXPI1 EXPI2 EXPI3 EXPI4 EXPI5 | 0.762 0.764 0.744 0.709 0.725 | 0.859 | 0.550 | |

To test discriminant validity, we applied the six-factor model to analyze and compare each model. As Table 2 shows, the fitting indices ($X^2$/DF = 1.305, RMSEA = 0.030, GIF = 0.914, NFI = 0.924, TLI = 0.978, IFI = 0.981, CFI = 0.981) in the six-factor model all meet the measuring standard, indicating the model is well fitted. Moreover, the fitting coefficients were better than the other models, indicating that the variables had good discriminant validity.

**Table 2.** The test results of variable discriminant validity (N = 279).

| Fit Indices | $X^2$/DF | RMSEA | GIF | NFI | TLI | IFI | CFI |
|---|---|---|---|---|---|---|---|
| Six-factor model: innovativeness, proactiveness, risk-taking, exploratory learning, exploitative learning, SIC of new ventures. | 1.305 | 0.030 | 0.914 | 0.924 | 0.978 | 0.981 | 0.981 |
| Five-factor model: innovativeness + proactiveness, risk-taking, exploratory learning, exploitative learning, SIC of new ventures. | 2.180 | 0.060 | 0.833 | 0.872 | 0.917 | 0.926 | 0.926 |
| Four-factor model: innovativeness + proactiveness + risk-taking, exploratory learning, exploitative learning, SIC of new ventures. | 3.091 | 0.080 | 0.782 | 0.816 | 0.852 | 0.867 | 0.866 |
| Three-factor model: innovativeness + proactiveness + risk-taking, exploratory learning + exploitative learning, and the SIC of new ventures. | 4.506 | 0.103 | 0.670 | 0.728 | 0.752 | 0.775 | 0.774 |
| Two-factor model: innovativeness + proactiveness + risk-taking, exploratory learning + exploitative learning + SIC of new ventures. | 5.676 | 0.119 | 0.608 | 0.655 | 0.669 | 0.697 | 0.695 |
| One-factor model: innovativeness + proactiveness + risk-taking + exploratory learning + exploitative learning + SIC of new ventures. | 6.022 | 0.124 | 0.602 | 0.632 | 0.645 | 0.673 | 0.671 |

We used Harman's single-factor test to check for common method variation while simultaneously conducting EFA of all items [64]. In the unrotated factor analysis results, the variance explanation rate of the first common factor was 36.037%. Therefore, common method deviation had a negligible effect on this study's results.

*4.2. Descriptive Analysis*

Pearson's correlation analysis of the study variables was conducted using SPSS24.0 software. Table 3 presents the results regarding the mean values, standard deviations, and

correlation coefficients. Results indicate a correlation between entrepreneurial orientation, ambidextrous learning, and the SIC of new ventures. Correlation coefficients were less than 0.75, indicating no multicollinearity problem among the variables [65,66]. Innovativeness, proactiveness, and risk-taking were positively correlated with the SIC of new ventures at r = 0.528 ($p < 0.01$), r = 0.513 ($p < 0.01$), and r = 0.435 ($p < 0.01$), respectively. Exploratory and exploitative learning were positively correlated with the SIC of new ventures at r = 0.537 ($p < 0.01$) and r = 0.548 ($p < 0.01$), respectively. Innovativeness, proactiveness, and risk-taking were positively correlated with exploratory learning at r = 0.522 ($p < 0.01$), r = 0.459 ($p < 0.01$), and r = 0.426 ($p < 0.01$), respectively. Innovativeness and proactiveness were positively correlated with exploitative learning at r = 0.452 ($p < 0.01$) and r = 0.547 ($p < 0.01$), respectively. The correlation coefficients between risk-taking and exploitative learning were relatively low.

**Table 3.** Descriptive statistics and correlation coefficient matrix for each variable (N = 279).

| Variable | Mean | SD | 1 | 2 | 3 | 4 | 5 | 6 | 7 | 8 |
|---|---|---|---|---|---|---|---|---|---|---|
| 1. Enterprise size | 2.767 | 0.863 | 1 | | | | | | | |
| 2. Enterprise age | 3.155 | 0.790 | 0.295 ** | 1 | | | | | | |
| 3. Innovativeness | 3.841 | 0.959 | 0.100 | 0.168 ** | 1 | | | | | |
| 4. Proactiveness | 3.843 | 0.978 | 0.127 * | 0.117 * | 0.415 ** | 1 | | | | |
| 5. Risk-taking | 3.367 | 1.100 | 0.014 | 0.122 * | 0.384 ** | 0.253 ** | 1 | | | |
| 6. Exploratory learning | 3.567 | 0.920 | 0.045 | 0.181 ** | 0.522 ** | 0.459 ** | 0.426 ** | 1 | | |
| 7. Exploitative learning | 3.841 | 0.801 | −0.004 | 0.08 | 0.452 ** | 0.547 ** | 0.284 ** | 0.420 ** | 1 | |
| 8. The SIC of new ventures | 3.627 | 1.022 | 0.051 | 0.159 ** | 0.528 ** | 0.513 ** | 0.435 ** | 0.537 ** | 0.548 ** | 1 |

Note: * $p < 0.05$, ** $p < 0.01$.

### 4.3. Hypothesis Testing

As Table 4 shows, the structural model used to test the research hypotheses can be considered a good fit for generalizing the results.

**Table 4.** Fit indices of the structural model (N = 279).

| Fit Indices | $X^2$/DF | RMSEA | GIF | NFI | TLI | IFI | CFI |
|---|---|---|---|---|---|---|---|
| Model 1: (EO-SIC) | 1.779 | 0.049 | 0.947 | 0.951 | 0.972 | 0.978 | 0.978 |
| Model 2: (EO-AL) | 1.641 | 0.044 | 0.929 | 0.930 | 0.966 | 0.972 | 0.971 |
| Model 3: (EO-AL-SIC) | 1.444 | 0.037 | 0.919 | 0.929 | 0.973 | 0.977 | 0.977 |
| Measuring standard | <3 | <0.08 | >0.9 | >0.9 | >0.9 | >0.9 | >0.9 |

Table 5 presents the SEM (Structural Equation Model) results, namely, the standardized path coefficients and probability levels of hypothesis testing. Results show that innovativeness (β = 0.330, $p < 0.05$), proactiveness (β = 0.344, $p < 0.05$), and risk-taking (β = 0.236, $p < 0.05$) had a significant and positive effect on the SIC of new ventures. Therefore, H1 is supported.

Moreover, innovativeness (β = 0.358, $p < 0.001$), proactiveness (β = 0.268, $p < 0.001$), and risk-taking (β = 0.233, $p < 0.001$) had a significant positive effect on exploratory learning. Innovativeness (β = 0.266, $p < 0.001$) and proactiveness (β = 0.490, $p < 0.001$) had a significant positive effect on exploitative learning. Therefore, hypotheses H2a, H2b, H2c, H3a, and H3b were supported. While the influence of risk-taking on exploitative learning was not significant (β = 0.064, $p > 0.05$), so hypothesis H3c was not supported. The results show that exploratory learning (β = 0.185, $p < 0.01$) had a positive effect on the SIC and exploitative learning (β = 0.268, $p < 0.001$) had a significant positive effect on the SIC of new ventures.

**Table 5.** The standardized path coefficients and probability levels of hypothesis testing.

| Model 1: (EO-SIC) | Paths | Estimate | SE | T-Value | *p*-Value | Results |
|---|---|---|---|---|---|---|
| | SIC←IT | 0.330 | 0.075 | 5.066 | *** | Supported |
| H1 (+) | SIC←PT | 0.344 | 0.067 | 5.802 | *** | Supported |
| | SIC←RT | 0.236 | 0.05 | 4.241 | *** | Supported |
| **Model 2: (EO-AL)** | | | | | | |
| H2a (+) | EXPR←IT | 0.358 | 0.067 | 5.193 | *** | Supported |
| H2b (+) | EXPR←PT | 0.268 | 0.065 | 4.385 | *** | Supported |
| H2c (+) | EXPR←RT | 0.233 | 0.053 | 3.995 | *** | Supported |
| H3a (+) | EXPI←IT | 0.266 | 0.055 | 3.775 | *** | Supported |
| H3b (+) | EXPI←PT | 0.490 | 0.059 | 7.102 | *** | Supported |
| H3c (+) | EXPI←RT | 0.064 | 0.043 | 1.08 | 0.28 | Not Supported |
| **Model 3: (EO-AL-SIC)** | | | | | | |
| H2a (+) | EXPR←IT | 0.358 | 0.067 | 5.203 | *** | Supported |
| H2b (+) | EXPR←PT | 0.268 | 0.066 | 4.384 | *** | Supported |
| H2c (+) | EXPR←RT | 0.234 | 0.053 | 4.003 | *** | Supported |
| H3a (+) | EXPI←IT | 0.265 | 0.055 | 3.777 | *** | Supported |
| H3b (+) | EXPI←PT | 0.490 | 0.059 | 7.102 | *** | Supported |
| H3c (+) | EXPI←RT | 0.065 | 0.043 | 1.082 | 0.279 | Not supported |
| H1a (+) | SIC←IT | 0.193 | 0.066 | 2.872 | 0.004 ** | Supported |
| H1b (+) | SIC←PT | 0.162 | 0.073 | 2.394 | 0.017 ** | Supported |
| H1c (+) | SIC←RT | 0.176 | 0.049 | 3.253 | 0.001 ** | Supported |
| H4a (+) | SIC←EXPR | 0.185 | 0.063 | 2.947 | 0.003 ** | Supported |
| H4b (+) | SIC←EXPI | 0.268 | 0.086 | 3.901 | *** | Supported |

Note: ** $p < 0.01$, *** $p < 0.001$.

We used the Sobel test to verify the significance of the mediating effect of ambidextrous learning on the relationship between EO and the sustainable innovation ability of new ventures. The Bootstrap number was set to 5000 [67]. According to the effect paths and confidence intervals shown in Table 6, exploratory learning has a significant mediating effect on the relationships between innovativeness and the SIC of new ventures ($\beta = 0.066$, $p < 0.05$), proactiveness and SIC ($\beta = 0.05$, $p < 0.05$), and risk-taking and SIC ($\beta = 0.043$, $p < 0.05$). Exploitative learning had a significant mediating effect on the relationship between innovativeness and sustainable innovation ability ($\beta = 0.071$, $p < 0.05$) and proactiveness and SIC ($\beta = 0.131$, $p < 0.05$), but not on risk-taking and SIC.

**Table 6.** Significance analysis of the mediating effect.

| Effect Path | Standardized Coefficients | Standard Error | 95% Confidence Interval (Lower) | 95% Confidence Interval (Upper) | *p*-Value |
|---|---|---|---|---|---|
| SIC←EXPR←IT | 0.066 | 0.032 | 0.014 | 0.143 | 0.015 |
| SIC←EXPR←PT | 0.050 | 0.028 | 0.012 | 0.125 | 0.012 |
| SIC←EXPR←RT | 0.043 | 0.02 | 0.008 | 0.091 | 0.014 |
| SIC←EXPI←IT | 0.071 | 0.034 | 0.02 | 0.158 | 0.003 |
| SIC←EXPI←PT | 0.131 | 0.051 | 0.054 | 0.252 | 0.002 |
| SIC←EXPI←RT | 0.017 | 0.017 | −0.014 | 0.057 | 0.246 |

To test the relationship between the equilibrium and interaction effects of ambidextrous learning and the SIC of a new venture, we conducted Linear Regression Analysis. As Table 7 shows, the explanatory rates of the equilibrium effect and interaction effect of ambidextrous learning on the SIC of new ventures are 43.1% and 42.5% respectively. The Durbin-Watson value is 1.988 and 1.955 respectively, close to 2, indicating a good Independence of residual. There was a significant linear relationship with enterprise size, enterprise age, exploratory learning, exploitative learning, and the equilibrium effects of ambidextrous learning on

the SIC of new ventures (F = 50.836, $p < 0.001$), and so is the interaction effect (F = 49.705, $p < 0.001$). The highest VIF value was 2.426 and 1.341 respectively, less than 10, and there were no multicollinearity problems among the independent variables. The equilibrium effect of ambidextrous learning had a significant positive effect on the SIC of new ventures ($\beta = 0.217$, $p < 0.01$), supporting H5. The interaction effect of ambidextrous learning had significant influence on the SIC of new ventures ($\beta = 0.131$, $p < 0.01$), supporting H6.

**Table 7.** Regression results of the equilibrium effect and interaction effect.

| Model Statistics | | Model 1 | Model 2 | Model 3 | Model 4 |
|---|---|---|---|---|---|
| Control variable | Enterprise size | 0.005 | 0.020 | 0.008 | 0.015 |
| | Enterprise age | 0.157 ** | 0.056 | 0.052 | 0.044 |
| Mediator variable | Exploratory learning | | 0.362 *** | 0.207 ** | 0.375 *** |
| | Exploitative learning | | 0.392 *** | 0.376 *** | 0.435 *** |
| Equilibrium effect of ambidextrous learning | | | | 0.217 ** | |
| Interaction effect of ambidextrous learning | | | | | 0.131 ** |
| $R^2$ | | 0.025 | 0.419 | 0.440 | 0.434 |
| Adjusted $R^2$ | | 0.019 | 0.412 | 0.431 | 0.425 |
| F | | 4.234 * | 58.694 *** | 50.836 *** | 49.705 *** |
| Maximum VIF | | 1.095 | 1.247 | 2.426 | 1.341 |
| Durbin–Watson value | | | | 1.988 | 1.955 |

Note: Dependent variable = the SIC of new ventures; N = 279; * $p < 0.05$, ** $p < 0.01$, *** $p < 0.001$.

## 5. Discussion and Conclusions

### 5.1. Discussion

The main purpose of this research was to define the SIC of new ventures from an ambidextrous perspective and discover the mediating role of ambidextrous learning on the relationship between EO and SIC of new ventures. The following research conclusions are supported by empirical data.

First, the three dimensions of entrepreneurial orientation have a significant positive impact on the SIC of new ventures, but the degree of impact is different. It is consistent with the findings of some previous studies, which are not ambidextrous perspective and not limited to the startup scene [37,40]. This result can be explained by the fact that EO drives new enterprises to take both hands when facing the disadvantage of new entry and produce ambidextrous tension in new ventures. On the one hand, EO drives new ventures to utilize resources more creatively and upgrade existing technologies, commodities, or businesses faster than their competitors. On the other hand, EO drives new ventures to pursue innovation in technology, products, and markets and prompts them to make quick decisions to adapt to external changes. That is to say, EO is an important means of solving the ambidextrous paradox of stable reuse and capability iteration and helps new ventures overcome organizational inertia and avoid the rigidity of capability. The results support the view that "EO is an invisible resource in new ventures and a powerful driving force to form SIC in new ventures" [18,38].

Second, ambidextrous learning partially mediates the relationship between EO and the SIC of new ventures. It is partially consistent with the findings of some previous studies [42,43] which focus on organizational learning. This result can be explained by the paradoxical tension between two seemingly opposing sides or Yin and Yang which is integral to sustainable development. Exploratory and exploitative learning reveal the inner path of EO to promote the formation of a new venture's sustainable innovation abilities. On the one hand, EO makes a new venture more willing to develop new products, services, and processes, creating greater demand for new knowledge, and encourages new ventures to seek out, research, and analyze novel information and ideas, identify new solutions to problems, and enter new fields of study. Exploratory learning emphasizes experimentation and innovation, breaks existing learning paths and conventions, and

promotes reorganizing resources and reforming organizational conventions. Doing so motivates the iteration and improvement of SIC and helps a venture avoid capability rigidity. On the other hand, limited by the disadvantages of new entry, new ventures pursue creative and comprehensive uses of existing knowledge resources, promoting mutual learning within and between different levels as well as the utilization, dissemination, and diffusion of knowledge. Exploitative learning emphasizes the understanding and utilization of existing knowledge and resources; values refinement, efficiency, and execution; expands the content and depth of knowledge resources, and deepens the understanding of organizational processes and conventions of new ventures. Such learning helps new ventures change their resource allocation and adapt to changes in the external environment, and promotes the precipitation and stability of SIC. The results support the view that "exploratory learning and exploitative learning are important pathways for enterprises to achieve sustainable innovation" [48].

Third, both the equilibrium and interaction effects of ambidextrous learning positively affect the SIC of new ventures. There has been little empirical research on this before [55,56]. This result lays a good foundation for further research on the relationship between exploratory learning and exploitative learning. Given the limited resources of new ventures, exploratory and exploitative learning compete for resources and have a crowding-out effect. Balanced ambidextrous learning implies that new ventures pay equal attention to both exploratory and exploitative learning. Maintaining an appropriate balance between exploring new possibilities and exploiting old certainties helps new ventures achieve both short-term survival and long-term development. The interaction of ambidextrous learning means synergistic effects between the two. Exploratory learning increases the depth of exploitative learning, while the depth, hierarchy, and effectiveness of exploitative learning contribute significantly to exploratory learning. Exploratory and exploitative learning behaviors efficiently influence and complement each other, which promote the construction of SIC. This results explains why new ventures not only pay attention to their own R&D investment, but also attach importance to external cooperation and the acquisition and utilization of existing resources.

*5.2. Theoretical Contributions*

SIC constitutes a new research field. Indeed, previous studies on sustainable innovation most simply emphasize "innovation over a considerable period of time", "continuous improvement", or equate it to traditional innovation. This is largely due to the difficulty of defining sustainability. This study significantly contributes to the literature by focusing on the SIC of new ventures from an ambidextrous perspective. Significantly, the fundamental difference between sustainable and traditional innovation is the tension generated by the innovation paradox. Ambidextrous tension is the core of sustainable innovation. A remarkable characteristic of the SIC of new ventures is the ambidexterity of stable reuse and capability iteration. Following this perspective, this study explored the prerequisites and influence paths of the SIC of new ventures.

This study also demonstrates that EO is a crucial strategic component for improving the SIC of new ventures. Scholars have hitherto regarded EO as the source of an enterprise's dynamic capability. However, it is unclear whether these studies are valid in the context of a new venture, whether this relationship is valid from an ambidextrous perspective, or what degree of influence it has. This study shows that EO is the key driving force behind the SIC of new ventures and verifies the direct and indirect effects of EO on the SIC of new ventures. EO can produce ambidextrous tension in new ventures. Accordingly, the results of this study serve to bridge organizational strategy and firm competence theory and provide explanatory models and analytical methods to explain the differences between the SIC of new ventures.

Moreover, the intermediate process in the formation and development from EO to SIC is unclear in the literature. This study reveals and verifies the partial mediating effect of ambidextrous learning between EO and SIC of new ventures. It also demonstrates that the

equilibrium and interaction effects of ambidextrous learning significantly influence the SIC of new ventures. Accordingly, this study provides data for unlocking the formation process of SIC in new ventures.

### 5.3. Managerial Implications

With respect to management practices, this study suggests three practical recommendations and implementation guidelines for new ventures in China. First, new ventures need to value EO, encourage creative thinking, and proactively grasp market opportunities. Some new ventures actively innovate when government policies encourage innovation and provide preferential policies and incentives. However, once the policy dividend disappears, innovation will also lose motivation and enthusiasm and will not be sustainable. So our study suggests that new ventures attach importance to an internal driving force, that is, EO. Founders and top managers highly value R&D, technology leadership, and innovation. They encourage employees to think independently, try new ideas and solutions, and advocate an innovative cultural atmosphere. New ventures proactively search for and identify business opportunities, pay close attention to market changes and customer needs, expand market boundaries, respond quickly, and take appropriate business or financial risks.

Second, new ventures could benefit from ambidextrous learning and maintain an interactive, cooperative, and dynamic balance between exploratory and exploitative learning. Limited by bounded rationality and resource constraints, new ventures regard exploratory and exploitative learning as contradictory and tend to favor one and ignore the other. So our study suggests that new ventures should not only pay attention to the huge short-term benefits brought by exploitative learning, but also allocate enough resources for exploratory learning. New ventures acquire strategic knowledge involving experimentation and high market risk. Meanwhile, they emphasize the use of existing product or service knowledge and experience. All these provide sufficient knowledge reserve for idea generation, idea transformation, and innovation output in new ventures.

Third, new ventures could benefit from the appropriate creation and activation of innovation tension, rather than the unilateral elimination of conflicts. In China, some enterprises avoid internal conflicts by following the "straight and narrow" as the tacit rule for survival under the influence of traditional cultural thinking, which is not conducive to the exchange of new ideas or the generation of new knowledge. Consequently, enterprises cannot effectively promote organizational learning behavior and innovation is inhibited. So our study suggests that new ventures could focus on gaining a dynamic understanding of innovation conflict to stimulate the vitality of enterprise innovation. Overly "peaceful" enterprises lack innovation vitality and moderate innovation conflicts promote healthy competition among organizations. New ventures should be more tolerant of differences, encourage the coexistence of diverse cultures and ideas, facilitate the exchange of views, cultivate employees' creativity and innovative thinking, and provide a constant source of knowledge for the sustainable innovation of new ventures.

### 5.4. Limitations and Future Directions

This study has several limitations. First, this study's research object was new ventures that have not entered the mature stage. However, as SIC development paths will differ according to industry, development stage, and resources, future studies should further subdivide new ventures. Second, the study area was restricted to areas with high innovation activity. Although this study employed a large sample, future studies should verify whether this study's findings are universal, explanatory, practical, and applicable to new ventures in other regions. Finally, this study selected ambidextrous learning as its mediating variable. However, other potential mediating variables with ambidextrous characteristics exist, such as knowledge transfer and integration. It is also possible for the equilibrium effect or interaction effect of ambidextrous learning. Further research is necessary to identify scientific analytical methods to test this possibility.

**Author Contributions:** Conceptualization, X.Y. and H.R.; methodology, X.Y.; software, X.Y. and N.C.; validation, X.Y.; formal analysis, X.Y.; investigation, X.Y. and H.R.; resources, X.Y. and N.C.; data curation, X.Y.; writing—original draft preparation, X.Y. and N.C.; writing—review and editing, X.Y.; visualization, X.Y. and N.C.; supervision, H.R.; project administration, X.Y. and N.C.; funding acquisition, X.Y. and H.R. All authors have read and agreed to the published version of the manuscript.

**Funding:** This research received no external funding.

**Institutional Review Board Statement:** Not applicable.

**Informed Consent Statement:** Informed consent was obtained from all subjects involved in the study.

**Data Availability Statement:** All data generated or analyzed during this study are included in the published article.

**Conflicts of Interest:** The authors declare no conflict of interest.

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
