# Peer review of "The Impact of Entrepreneurial Orientation on the Sustainable Innovation Capabilities of New Ventures: From the Perspective of Ambidextrous Learning"

_sustainability, doi:10.3390/su15119026_

Round 1

Reviewer 1 Report

Introduction includes all the necessary information, however I would like to see some more literature and sources to support your research aims.

The cunclusion section does not combines research outcomes. It should be more specific

The paper needs more suggestions

 More terminology of Entrepreneurial Orientation of English language required

Reviewer 2 Report

Suggestion: properly focus the summary clearly showing objectives, methodology, results and main conclusion.

 Suggestion: include an opening paragraph between sections 2. and 2.1. Sugestions: the introduction is too long. Authors could focus text in order to make it more direct and short, as well as accurately oriented to the topic(s) the study deals with.  Suggestion: the texts in general are, in my opinion, a bit long. They should be shortened and more accurately oriented to the respective topic covered in each section.  Suggestion: include an opening paragraph between sections 3. and 3.1.  Suggestion: verify that the number of questionnaires collected allows answering the N/P criterion of number of cases with respect to the number of questions that the relevant theory sets at 10.  Suggestion: it is necessary that the authors incorporate in detail the references that support the methods and data processing used in this work. Sugestions: interpret in a little more detail each of the tables presented in the work. Sugestions: interpret in a little more detail each of the tables presented in the work. Especially some of the results that might not confirm the results (Table 2).  Sugestions: interpret in a little more detail each of the tables presented in the work. Especially some of the results that might not confirm the results (Table 5). Sugestions:interpret in a little more detail each of the tables presented in the work. Especially some of the results that might not confirm the results (Table 7).  Suggestion: It is absolutely necessary that in the discussion each of the results be contrasted with the findings available in the literature.  Suggestion: it is necessary that the conclusions are accurate to respond to the objectives, questions and hypotheses stated in this work.  Suggestion: the conclusions should only conclude, it does not seem appropriate to continue analyzing results. Suggestion: the number of references seems excessive (64), these should be precise and the most relevant and updated to support the work presented. The writing is clear and understandable. I have no observations.

Reviewer 3 Report

Thank you for the opportunity to review this manuscript on the impact of entrepreneurial orientation on sustainable innovation capability. The topic of the study is interesting and important.

In the following I address some notions and questions that I have concerning the manuscript. I'd like to start with the research questions that are quite many. Almost all of the three numbered RQs includes multiple questions. In addition, latter questions assume an answer to the first two questions. Therefore, I'd encourage the authors to return to the formulation of the RQs and consider how to formulate the questions.

Second, is the definition of entrepreneurial orientation (EO). Based on the theoretical discussion, it remains unclear to me whether EO is examined as an organizational level or an individual level concept in the study. Also the definition of new venture is somewhat confusing in terms of the level of analysis. Being more explicit on the level of analysis would help the reader to follow the line of argumentation.

In section 2.3 it is written: "SIC in new ventures involves taking market opportunities as the starting point". I was wondering the approach the study takes into entrepreneurial orientation (and dynamic capabilities). This quoted section seems to articulate a reactive approach. Have you considered the proactive side of creating opportunities? 

Section 2.4. introduces a concept on "dynamic core capability". Could you please elaborate what is meant with this concept, since it is not quite established in the literature?

The hypotheses do no follow the same logic throughout, as H2 and H3 have both the main hypothesis and sub-hypotheses, but H4 and H5 do not have similar structure despite being divided into sub-hypotheses. Would it be possible to structure the hypotheses consistently?

Could you please be more explicit on the selection criteria: "had not entered the mature stage according to enterprise life cycle theory". What does this mean in practise? How was it ensured?

Section 3.2 lists that the study measured the "sustainable capability of new ventures" Is there an error or do you introduce a new concept here?

H3c not being supported does not seem like a surprise when looking at the statements through which risk-taking is measured. 

Finally, I'd like to ask about the fit of the paper for this journal. The way I read the paper, sustainable innovation refers to continued innovation in the study. The paper does not address ecological or social aspects of sustainability, it's focused purely on financial sustainability. To me this seems like an issue, but I'd like to learn about the authors' perspective on why they see a fit.
